# Dietary intake and associated risk factors among pregnant women in Mbeya, Tanzania

**Erick Killel**[1], **Geofrey Mchau**[1], **Hamida Mbilikila**[1], **Kaunara Azizi**[1], **Nyamizi Ngasa**[1], **Adam Hancy**[1], **Tedson Lukindo**[1], **Ramadhan Mwiru**[2], **Ramadhan Noor**[2], **Abraham Sanga**[2], **Patrick Codjia**[2], **Germana H. Leyna**[1,3], **Ray M. Masumo**[1,4]*

1 Department of Community Health and Nutrition, Tanzania Food and Nutrition Centre, Dar es Salaam, Tanzania, 2 The United Nations Children's Fund (UNICEF), Dar es Salaam, Tanzania, 3 Department of Epidemiology and Biostatistics, Muhimbili University of Health and Allied Sciences, Dar es Salaam, Tanzania, 4 Department of Statistics, University of Dar es Salaam (UDSM), Dar es Salaam, Tanzania

* rmasumo@yahoo.com

**Data Availability Statement:** All datasets underlying this study are freely available at the public repository https://osf.io/7ysb9/.

## Abstract

Poor dietary intake among pregnant women has serious detrimental consequences for pregnancy and offspring both in developed and developing countries. This study aimed to assess dietary intake and associated risk factors among pregnant women. A cross-sectional study was conducted in Mbeya, Tanzania with a sample size of 420 pregnant women attending antenatal clinics to assess the factors associated with dietary intake. Dietary intake was assessed using a piloted questionnaire of the Prime Diet Quality Score. A tested standard questionnaire was also used to collect factors that are associated with dietary intake among pregnant women. The strengths of the associations between the dependent and independent variables were tested using the Pearson chi-square tests and the multivariate log-binomial regression method was performed to calculate the adjusted risk ratios (ARR) and 95% confidence interval (CI). The study revealed that out of 420 pregnant women who participated in this study only 12.6% and 29.3% consumed at least four servings of fruits and vegetables per week respectively. Poor dietary intakes were less likely among cohabiting pregnant women [Adjusted RR 0.22 (95% CI 0.09–0.50)] and; those who reported taking Fansidar tablets during the pregnancy [Adjusted RR 0.55 (95% CI 0.31–0.96)]. Further, we found that poor dietary intakes were more likely among pregnant women who were classified as overweight and obesity by the MUAC above 33cm [Adjusted RR 3.49 (95% CI 1.10–11.06)]. The study results affirm that cohabitation and obesity affect dietary intakes among pregnant women differently compared to married women in rural settings of Tanzania. Further research is needed to investigate the social aspects that link dietary intake outcomes for developing a tailored gestational intervention to improve maternal and birth outcomes in sub-Saharan African countries.

## Introduction

Poor dietary intake among pregnant women has serious detrimental consequences for pregnancy and offspring both in developed and developing countries [1]. In 2017, Pelletier and

**Funding:** The authors have declared that no competing interests exist.

**Competing interests:** The authors received no specific funding for this work.

colleagues defined dietary quality as one that is hygienically safe, nutritious, balanced, and well adapted to the needs of individuals in order to prevent disease, ensure a good state of health, as well as proper development [2]. Previous studies have consistently reported that the Mediterranean diet is one of the most effective diets in reducing the risk of cardiovascular diseases and overall mortality due to non-communicable diseases [3, 4]. The Mediterranean diet is characterized by a high intake of fish, olive oil, non-starchy vegetables, legumes, whole grains (cereals), fruits, and nuts, as well as a lower intake of dairy products, red and processed meat and a moderate intake of wine [3, 4]. Worldwide dietary guidelines vary and each country adapts to suit its specific needs [5, 6]. The dietary guidelines of the United Kingdom and the American 2020–2025 recommended the consumption of at least two portions of fruit and 3 portions of vegetables a day [7, 8].

A Lancet study conducted in 195 countries on the health effects of dietary risks published that in most of the countries, the intake of healthy food such as whole grains, vegetables and fruits were much less compared to unhealthy foods such as processed foods and soft drinks [5]. An epidemiological study from South Africa by Venter and Winterbach revealed a higher dietary intake of fats than the recommended among mid-adolescents [9]. Evidence emanated from Bahrain consistently reported poor consumption of healthy food items compared to unhealthy food items [10]. There are a growing number of studies on dietary intake among pregnant women in many countries, especially industrialised countries [5, 6], however, published research regarding dietary intake among pregnant women in Tanzania is minimal. While the results from studies based in other countries provide relevant information related to this subject [9], these results cannot be entirely relatable to pregnant women in Tanzania. A recently published longitudinal study in Dar es Salaam, Tanzania among pregnant women reported high consumption of green leafy vegetables and refined grains [11]. Inconsistent findings were reported in another prospective cohort study among 432 pregnant women in the rural settings of Ethiopia where the consumption of vegetables and fruits was poor and associated with a higher risk of adverse pregnancy outcomes [12]. The research work in Tanzania [11] was performed in urban settings and may not represent the dietary intake in rural settings. The two previous National Nutrition Surveys in Tanzania, TNNS of 2014 and 2018 lack detailed information on dietary intake and diet quality [13]. Therefore, further high-quality prospective cohort studies are required in Tanzania to enhance the generalisability of the results and help inform policies and programmes.

Dietary intake among pregnant women is affected by various factors, such as socio-demographic and economic status, nutrition status, environmental, cultural, and political [14–18]. Adequate dietary intake is nothing new to sub-Saharan African countries but what is of great concern is the fact that it is one of the issues on which a lot of resources have been spent over a period of time with very limited results. The current pieces of literature provide limited information on dietary intake among pregnant women in sub-Saharan African countries because the requirements that would enhance the collection and use of those data, including the use of new technology, in these countries rarely exist [19]. Hence, the present study aimed at examining dietary intake and associated risk factors among pregnant women in the Mbeya region, Tanzania. The findings of this study would be a valuable step in developing a tailored gestational intervention to improve maternal and birth outcomes in sub-Saharan African countries.

## Methods

### Ethics statement

The survey was approved by the Tanzania Ethics Committees i.e. the National Institute for Medical Research with the reference number SZEC-24239/R.A/V.1/151. Date of issue 12[th]

August 2022. All eligible subjects were informed of the purpose and nature of the survey and those who agreed to participate were asked to sign a written informed consent form. Moreover, a written informed consent was obtained from the parent/guardian of each participant under 18 years of age. All procedures followed were per the ethical standards of the Helsinki Declaration of 1975 including the confidentiality and, authors had no access to information that could identify individual participants during or after data collection.

### Study design

A cross-sectional study was conducted among pregnant women in seven districts of Mbeya region in Tanzania. The study was carried out from 15[th] September to 10[th] December 2022.

### Study area

Mbeya region has a population of 2,204,543 (1,068,615 male and 1,135,928 female) and 557,574 women of reproductive age [20]. The total deliveries in the Mbeya region in 2020 were 72,076. There are 17 hospitals, 23 health centres and 278 dispensaries, where 251 health facilities provide reproductive and child health services. This study was conducted at 42 Reproductive and Child Health (RCH) Clinics in seven districts of the Mbeya region. The selected RCH clinics in this study are estimated to provide services to approximately 1036 pregnant women [20].

### Study population

All pregnant women aged between 15 to 49 years, less than 28 weeks of gestation, and who attended antenatal visits in Mbeya were invited to participate in the study. This is according to the minimal risk in research involving pregnant women and offspring [21]. A total of 574 pregnant women were invited and 420 (response rate of 73.0%) agreed to participate. The study excluded pregnant women taking medication for other reasons except malaria chemoprophylaxis plus iron and folate supplements.

### Sample size and sampling procedure

A sample size (n = 420) was considered sufficient based on the Lwanga and Lemeshow formula [22]. Prior to carrying out the study, the proportion of women of reproductive age with poor dietary intake was estimated to be 45%, with a margin error of 5%, a confidence level of 95%, and a design effect of 1.5. Another 10% was added to the sample size to account for non-responses. The sampling procedure involved two steps [22], a list of 251 governmental and faith-based health facilities providing antenatal services in the Mbeya region was obtained and used in a random selection of the health facilities from each district based on probability proportional to size sampling. A total of 42 facilities from a pool of 251 were randomly selected for the survey. An additional two reserved clusters were included in the survey. Given the sampling frame of public health facilities in Mbeya, the probability proportional to size was performed to allocate the number of facilities per district for inclusion in the survey. Therefore, a total of 44 health facilities offering antenatal services located in the Mbeya region were visited and surveyed [22].

### Data collection

**Dietary intake assessment.** Dietary intake was assessed by the Prime Diet Quality Score (PDQS) developed in the USA using a modified Prime Screen questionnaire as a means to characterize diet quality [23]. The questionnaire was first found to predict factors associated

with the lower risk of coronary heart disease (CHD) in a large population in the USA [23], and diet quality among adults in Bosnia and Herzegovina [18]. In Tanzania, Yang and colleagues employed the questionnaire in a prospective pregnancy cohort study [11]. The PDQS contains 21 food groups; 13 are healthy food and, seven are unhealthy food. The PDQS was assessed using 24-hour recalls, which reflected the feeding practice from the previous morning to the morning of the interview [11, 23, 24]. For this study, the questionnaire was translated into Kiswahili the main language in Tanzania, spoken proficiently by almost 95% of the population. In the translation process, two translators with different backgrounds independently translated the original questionnaire into Kiswahili. The IMAN project staff in the field reviewed for semantic, experiential, and conceptual equivalence to the original version. Sensitivity to culture and selection of appropriate words were considered. The Kiswahili version of the questionnaire was then given to a translator fluent in both English and Kiswahili to translate back into the original language. This translator was not shown the original English version. Lastly, all translations and the original questionnaire were given to IMAN project staff in the field in order to consolidate all the versions of the questionnaire and achieve equivalence between the original and target versions. Both the Kiswahili version and the original English version of the PDQS were administered to 20 female secondary school teachers in Dar es Salaam in two sessions separated by an interval of two weeks to evaluate the quality of the translations in terms of comprehensibility, readability and relevance on face validity and, correlations between the two administrations were calculated. However, a 30-day recall was not a part of this study which might have different comparable outcomes.

Participants were asked 'From when you woke up yesterday till you woke up this morning did you consume the following food items: dark green leafy vegetables, cruciferous vegetables, dark orange vegetables and fruits, other vegetables, citrus fruits, other fruits, legumes, nuts and seeds, poultry, fish, whole grains, vegetable liquid oils, white roots and tubers, red meat as a main dish, processed meats, refined grains and baked products, sugar-sweetened beverages, fried foods away from home, sweets, ice cream and low-fat dairy?' Responses were given on a 5-point likert scale; 0 = not at all, 1 = once, 2 = twice, 3 = thrice, and 4 = fourth or more. Each occasion of food group consumption was considered as a serving. The mean number of servings was computed over the available recall days for each participant. The mean number of servings for each food group was multiplied by 7 to standardise the number of servings per week, from which points for each food group could be assigned based on whether the food was categorised as healthy or unhealthy [11, 23, 24]. Points were assigned for consumption of healthy food groups as follows: 0–1 serving/week, 0 points; 2–3 servings/week, 1 point; and ≥4 servings/week, 2 points. Scoring for unhealthy food groups was assigned as follows: 0–1 serving/week, 2 points; 2–3 servings/week, 1 point; and ≥4 servings/week, 0 points [11, 23, 24].

**Demographic and socio-economic factors.** The demographic factors were assessed by asking pregnant women attending antenatal services to provide the following information: Age, marital status, education level, and occupation status. Socio-economic status was assessed by household ownership of durable assets (such as ownership of a car, motorcycle, bicycle, cart, refrigerator, television, radio, etc.), housing characteristics (such as the material of dwelling floor and roof, toilet facilities, etc.), and access to basic services (such as electricity supply, source of drinking water). Household asset data uses simple questions and therefore suffers from less recall or social desirability bias.

**Anthropometric measurements.** Maternal nutrition status was assessed by measuring weight and height. Weight was measured by the nearest 0.1 kg with a battery-powered electronic scale (Seca, Hamburg, Germany), and height was measured to the nearest 0.1 cm with a height model recommended by UNICEF. Height was measured when pregnant women were not wearing shoes or a head covering. Further, Mid Upper Arm Circumference

(MUAC) assessed by MUAC tapes was used to assess the nutrition status of pregnant women [25].

**Laboratory assessment.**   A trained nurse collected blood samples through vein puncture from consented participants. Blood samples were taken into ethylenediaminetetraacetic acid (EDTA) and non-anticoagulated whole blood vacutainers (Becton Dickenson, NJ, USA). Approximately 6mL of venous blood sample was collected on each vacutainer and protected from light. Whole-blood vacutainers were maintained at 4–8˚C for less than 2 hours before being transported to the temporary laboratories. Malaria was tested by rapid diagnostic test (SD Bioline, Rep. of Korea), and hemoglobin level was measured by HemoCue HB 201+ analyser (Hemo Cue, Angelholm, Sweden). Assessment of C-reactive protein (CRP), and alpha-1 acid glycoprotein (AGP) was performed with Roche Cobas Integra 400 Plus analyser (Roche Diagnostics GmbH, German). Hemoglobin levels <12.0 and <8.0 g/dL were used to characterise anaemia and severe anaemia, respectively. Serum C- reactive protein (CRP) and Alpha-1-acid glycoprotein (AGP) values of CRP > 5.0 mg/L and AGP > 1.0 g/L respectively were characterized as high inflammatory marks [26, 27].

## Data analysis

The data were analysed by using SPSS version 25. The dietary intake as dependent variable was assessed as a categorical variable splitting at the median i.e. 0 = good dietary intake and, 1 = poor dietary intake.

An asset-based approach to measuring household socio-economic status is considered an alternative to income and consumption expenditure in low-income countries. Principal Components Analysis (PCA) is a method for determining wealth indices [28]. In this study, household wealth index was assessed as (1) "available and in working condition" or (0) "not available and/or not in working condition" of durable assets, housing characteristics and access to basic services. For constructing a wealth index among pregnant women in Mbeya, the first principal component was used to categorise households into two approximate group's i.e. lowest and highest group. The strengths of the associations between the dependent and independent variables in bivariate analysis were tested using the Pearson chi-square tests because all variables were categorical. Independent variables that were significant at arbitrary levels in the bivariate analysis were selected for multivariate analysis. We based this on the Wald test with a P-value cut-off of 0.7. In multivariate analysis, Log binomial regression method were used first for adjusting confounders and second to identify independent predictors of dietary intake among the study population, and the significance level was set at 5%.

## Results

### Reliability

The internal consistency reliability scales were examined using Cronbach's alpha. Test-retest reliability analysis was performed using kappa statistics and Intra class correlation coefficients (ICC). The agreement between the interviewers and the gold standard on the dietary intake assessments on the English and Kiswahili versions were Cohen's kappa of 0.62 and 0.67 respectively. During the field, duplicate interviews were performed randomly with 20 pregnant women. Test-retest reliability of reports on the dietary intake assessment using a Kiswahili version in terms of ICC was 0.72 (95% CI 0.64–0.78). Thus, acceptable levels of intra-interviews agreement (kappa >0.60) were obtained [29].

## The characteristics of the study population

Study participants had a mean age of 25.49 ± 6.37 years. Table 1 depicts the characteristics of the study population. More than half of the participants were 15–24 years old, about seventy-two percent had completed at least primary education and, 84.3% were self-employed. Five percent (n = 21) of pregnant women had Mid Upper Arm Circumference (MUAC) of above 33cm and falls in the category of overweight and obesity. Nine percent (n = 38) of pregnant women had serum C- reactive protein (CRP) above 5mg/L and, 19% (n = 80) had Alpha-1-acid glycoprotein (AGP) above 1 g/L. For the construction of wealth index: One-third of the participants lived in houses with electricity; 72.0% had access to improved sources of drinking water and 65.0% were not sharing toilet facilities. The pit latrine without washable was the most common type of toilet 31.6% (n = 133). About half (51.5%) used cement as the material of the dwelling floor and 7.8% used thatch/palm leaf as material for the roof. Furthermore, less

**Table 1. Frequency distribution of the socio-demographic and economic characteristics of pregnant women in Mbeya (n = 420).**

| Variable | Categories | Frequency (n) | Percentage (%) |
|---|---|---|---|
| Age group (Years) | 15–19 | 82 | 19.5 |
| | 20–24 | 133 | 31.6 |
| | 25–29 | 99 | 23.6 |
| | 35+ | 106 | 25.2 |
| Education | No education | 34 | 8.1 |
| | Primary | 301 | 71.7 |
| | Secondary and above | 85 | 20.2 |
| Marital status | Married | 238 | 56.5 |
| | Cohabit | 133 | 31.6 |
| | Single/ Divorced | 49 | 11.6 |
| Number of pregnancies | Primigravida | 104 | 24.7 |
| | Multigravida | 316 | 75.1 |
| Trimester of pregnancy | First trimester (<12 weeks) | 109 | 26.0 |
| | Second trimester (12–26 weeks) | 311 | 74.0 |
| Received of iron and folic acid supplements | No | 155 | 36.8 |
| | Yes | 265 | 62.9 |
| Received Fansidar (SP) during pregnancy | No | 204 | 48.5 |
| | Yes | 216 | 51.3 |
| Occupation status | Formal employment | 15 | 3.6 |
| | Self employed | 355 | 84.3 |
| | Not employed | 51 | 12.1 |
| Household wealth Index | Higher socio-economic | 140 | 33.4 |
| | Middle socio-economic | 139 | 33.2 |
| | lower socio-economic | 140 | 33.4 |
| Mid Upper Arm Circumference (MUAC) | Thin (<23cm) | 16 | 3.8 |
| | Normal (between 23 and 33cm) | 383 | 91.2 |
| | Overweight or obesity (above 33 cm) | 21 | 5.0 |
| Malaria infection | No | 402 | 95.7 |
| | Yes | 18 | 4.3 |
| Serum C- reactive protein (CRP) | CRP≤5mg/L | 383 | 91.0 |
| | CRP>5mg/L | 38 | 9.0 |
| Alpha-1-acid glycoprotein (AGP) | AGP< = 1 g/L | 341 | 81.0 |
| | AGP>1 g/L | 80 | 19.0 |

**Table 2. The patterns and distribution of food groups consumption among pregnant women according to PDQS score (n = 420).**

| Healthy foods | | | |
|---|---|---|---|
| *Serving per week* | *0–1 serving/week n (%)* | *2–3 servings/week n (%)* | *≥4 servings/week n (%)* |
| Dark leafy green vegetables | 142(33.7) | 156 (37.1) | 123 (29.2) |
| Cruciferous vegetables | 393(93.4) | 20(4.8) | 8(1.9) |
| Dark orange vegetables and fruits | 263(62.5) | 105(24.9) | 53(12.6) |
| Other vegetables | 266(63.2) | 93 (22.1) | 62(14.7) |
| Whole citrus fruits | 391 (92.9) | 25 (5.9) | 5 (1.2) |
| Other whole fruits | 308(73.2) | 78(18.5) | 35(8.3) |
| Legumes | 259 (61.5) | 104 (24.7) | 58(13.9) |
| Nuts and seeds | 275(65.3) | 107(25.4) | 39(9.3) |
| Poultry | 389(92.4) | 21(5.0) | 11(2.6) |
| Fish | 277(65.8) | 98(23.3) | 46(10.9) |
| Whole grains | 324(77.0) | 68(16.1) | 29(6.9) |
| Vegetable liquid oils | 46(10.9) | 134(31.8) | 241(57.2) |
| White roots and tubers | 189(44.9) | 161(38.2) | 71(16.9) |
| Low fat diary | 349(82.9) | 56(13.3) | 16(3.8) |
| **Unhealthy foods** | | | |
| *Serving per week* | *0–1 serving/week n (%)* | *2–3 servings/week n (%)* | *≥4 servings/week n (%)* |
| Red meats | 311(73.9) | 85(20.2) | 25(5.9) |
| Sweets and ice cream | 349(82.9) | 58 (13.8) | 14 (3.3) |
| Fried foods obtained away from Home | 348(82.7) | 65(15.4) | 8(1.9) |
| Processed meat | 412(97.9) | 8(1.9) | 1(0.2) |
| Refined grains and baked goods | 74(17.6) | 145(34.4) | 202 (48.0) |
| Sugar sweetened beverages | 249(59.1) | 146(34.7) | 26(6.2) |

than two percent (n = 6) and 2.4% (n = 10) of the participants owned a motor vehicle and a set of television respectively (not in Table 1). Following the PCA analysis, 140 pregnant women (33.4%) fell under the category of lower socio-economic status as shown in Table 1.

## Dietary intake and diet quality

Among 420 pregnant women who participated in the study, two hundred forty (57.2%) fell into the group of poor dietary intake. The median PDQS was 16 (the 25th and 75th percentiles were 14.0 and 18.0, respectively). Table 2 shows the patterns and distribution of dietary intake among pregnant women according to PDQS on healthy and unhealthy food shows that 57.2 of the study participants consumed more than four servings of edible vegetable liquid oil per week out of the 14 healthy foods assessed. Furthermore, the healthy foods that were less consumed per week were cruciferous vegetables (93.4%), whole citrus fruits (92.9%), and poultry (92.4%). However, refined grains and baked goods represented the highest percentage of servings consumed per week out of the six unhealthy foods assessed.

## Bivariate analysis

Table 3 depicts the bivariate analysis; poor dietary intake were significantly associated with the marital status of pregnant women, and those who received Fansidar tablets during pregnancy (p>0.05). However, the age group of pregnant women, their educational level, occupation

**Table 3. Bivariate analysis on the factors associated with poor dietary quality among pregnant women in Mbeya (n = 420): Chi square test.**

| Variable | Categories | Good dietary quality % (n) | Poor dietary quality % (n) | P-value |
|---|---|---|---|---|
| Age group (Years) | 15–19 | 16.8 (30) | 21.6 (52) | 0.129 |
|  | 20–24 | 30.7 (55) | 32.4 (78) |  |
|  | 25–29 | 29.1 (52) | 19.5 (47) |  |
|  | 35+ | 23.5 (42) | 26.6 (64) |  |
| Education | No education | 8.3 (15) | 7.9 (19) | 0.696 |
|  | Primary | 70.6 (127) | 72.6 (175) |  |
|  | Secondary and above | 21.1 (38) | 19.5 (47) |  |
| Marital status | Married | 70.0 (126) | 46.5 (112) | 0.000 |
|  | Cohabit | 17.8 (32) | 42.3 (102) |  |
|  | Single/ Divorced | 12.2 (22) | 11.2 (27) |  |
| Trimester of pregnancy | First trimester (<12 weeks) | 27.8 (50) | 24.9 (60) | 0.506 |
|  | Second trimester (12–26 weeks) | 72,2 (130) | 75.1 (181) |  |
| Received of iron and folic acid supplements | No | 38.5 (69) | 35.7 (86) | 0.548 |
|  | Yes | 61.5 (110) | 64.3(155) |  |
| Received Fansidar during pregnancy | No | 43.6 (78) | 52.3 (126) | 0.077 |
|  | Yes | 56.4 (101) | 47.7 (115) |  |
| Occupation status | Formal employment | 5.6 (10) | 2.1 (5) | 0.156 |
|  | Self employed | 83.3 (150) | 85.5 (206) |  |
|  | Not employed | 11.1 (20) | 12.4 (30) |  |
| Household wealth Index | Higher socio-economic | 33.0 (59) | 34.0 (82) | 0.365 |
|  | Middle socio-economic | 30.2 (54) | 35.3 (85) |  |
|  | lower socio-economic | 36.9 (66) | 30.7 (74) |  |
| Mid Upper Arm Circumference (MUAC) | Thin (<23cm) | 3.9 (7) | 3.7 (9) | 0.079 |
|  | Normal (between 23 and 33cm) | 91.6 (164) | 90.9 (219) |  |
|  | Overweight or obesity (above 33 cm) | 4.5 (8) | 5.4 (13) |  |
| Malaria infection | No | 96.7 (174) | 95.0 (229) | 0.409 |
|  | Yes | 3.3 (6) | 5.0 (12) |  |
| Serum C- reactive protein (CRP) | CRP≤5mg/L | 89.4 (161) | 92.1 (222) | 0.344 |
|  | CRP >5mg/L | 50.0 (19) | 7.9 (19) |  |
| Alpha-1-acid glycoprotein (AGP) | AGP< = 1 g/L | 80.6 (145) | 81.3 (196) | 0.542 |
|  | AGP>1 g/L | 19.4 (35) | 18.7 (45) |  |

status, household wealth index, received iron and folic acid supplements, Serum C- reactive protein (CRP) and Alpha-1-acid glycoprotein (AGP) were not significantly associated with poor dietary intake and diet quality among pregnant women (p>0.05).

## Multivariate analysis

Table 4 presented all variables which were significant at arbitrary levels in the bivariate analysis and hence qualify to be included in the multivariate analysis. Using a Log binomial regression method, the study found out that poor dietary intake were less likely among cohabiting pregnant women [Adjusted RR 0.22 (95% CI 0.09–0.50)] and; those who reported taking Fansidar (Sulfadoxine and Pyrimethamine, SP) tablets during pregnancy in Mbeya region [Adjusted RR 0.55 (95% CI 0.31–0.96)]. Further, the study found that poor dietary intake were more likely among pregnant women who were classified as overweight and obesity by the MUAC [Adjusted RR 3.49 (95% CI 1.10–11.06)] and; slightly significant among pregnant women of

**Table 4. Multivariate log binomial regression methods were used to assess factors associated with poor dietary intake among pregnant women in Mbeya (n = 420).**

| Variable | Category | Adjusted RR | 95% confidence interval for Adj. RR | |
|---|---|---|---|---|
| | | | Lower bound | Upper bound |
| **Age group** | 15–19 | 1.22 | 0.51 | 2.89 |
| | 20–24 | 1.31 | 0.66 | 2.59 |
| | 25–29 | **2.05** | **0.98** | **4.29** |
| | 35+ | 1 | 1 | 1 |
| Marital status | Married | 1.35 | 0.60 | 3.04 |
| | Cohabit | **0.22** | **0.09** | **0.50** |
| | Single/divorced | 1 | 1 | 1 |
| Occupational status | Formal employment | 4.29 | 0.83 | 22.17 |
| | Self employed | 1.02 | 0.45 | 2.29 |
| | Not employed | 1 | 1 | 1 |
| Education level | No formal | 0.73 | 0.21 | 2.56 |
| | Primary | 0.95 | 0.47 | 1.91 |
| | Secondary and above | 1 | 1 | 1 |
| Household wealth index | Higher socio-economic | 1.40 | 0.77 | 2.56 |
| | Middle socio-economic | 0.70 | 0.36 | 1.35 |
| | lower socio-economic | 1 | 1 | 1 |
| Mid Upper Arm Circumference (MUAC) | Thin (<23cm) | 3.02 | 0.52 | 17.50 |
| | Overweight or obesity (above 33 cm) | **3.49** | **1.10** | **11.06** |
| | Normal (between 23 and 33cm) | 1 | 1 | 1 |
| Taken Fansidar during this pregnancy | Yes | **0.55** | **0.31** | **0.96** |
| | No | 1 | 1 | 1 |
| Malaria status | Yes | 1.89 | 0.54 | 6.62 |
| | No | 1 | 1 | 1 |
| Received of iron and folic acid supplements | Yes | 1.09 | 0.62 | 1.90 |
| | No | 1 | 1 | 1 |
| Alpha-1-acid glycoprotein (AGP) | AGP< = 1 g/L | 0.94 | 0.48 | 1.81 |
| | AGP>1 g/L | 1 | 1 | 1 |
| Serum C- reactive protein (CRP) | CRP≤5mg/L | 0.60 | 0.24 | 1.50 |
| | CRP>5mg/L | 1 | 1 | 1 |
| Number of pregnancy | Primigravida | 1.09 | 0.54 | 2.22 |
| | Multigravida | 1 | 1 | 1 |
| Trimester of pregnancy | First trimester (<12 weeks) | 1.14 | 0.64 | 2.03 |
| | Second trimester (12–26 weeks) | 1 | 1 | 1 |

The reference group is last category

age group 25–29 years old [Adjusted RR 2.05 (95% CI 0.98–4.29)]. Poor dietary intake was not associated with higher concentrations of inflammatory factors i.e. CRP and AGP.

## Discussion

This study contributes to our understanding on socio-demographic drivers for poor dietary intake among pregnant women legally married or cohabiting in Tanzania. Worldwide, poor dietary intake has negative consequences for pregnancy and born children [1, 12]. This study found that dietary intake among pregnant women in the rural settings of Tanzania was largely characterised by low intakes of fruits and vegetables. The findings are very similar to those

found in Ethiopia [12], Iran [30], and India [31]. However, a study from the urban settings of Tanzania reported a high consumption of vegetables among pregnant women [11]. This could be explained by the difference in research methodology especially the inclusion and exclusion criteria of the study participants, and also the difference in dietary assessment tools used between the studies. Evidence shows that, the nutrients in fruits and vegetables such as fibers, vitamins, minerals, and phytochemicals play a key role in human health and well-being [32–35]. Tanzania has recently developed its national food based dietary guidelines and, it has not yet been fully operationalized. Guidelines of other countries like UK and USA recommend a plant-based diet, rich in fruit, vegetables, whole grains, and legumes to lower the risk of heart disease, type 2 diabetes, obesity, and other health conditions [8, 27].

In this study, we highlight the factors that affect the dietary intake of pregnant women in the rural settings of Tanzania. Our findings affirmed previous researches on the relationship between cohabitation and dietary intake among pregnant women [36]. Pregnant women who engaged in cohabiting relationships seemed to have better education and financial position and, thus reflected on their decision power over their wealth [36]. From the anecdotal evidence, it is known that cohabiting relationships are a common practice in the study area. According to the study of Dinour and colleagues in 2012, they found that marital status is one of strong socio-demographic factors that greatly influence health-related behaviours and outcomes [37]. However, studies emanating from sub-Saharan African countries merge cohabitation and marriage into one category of marital status and, this is because pregnant women are reluctant to report the status of cohabitation because of stigma [38]. This study managed to assess separate the marriage and cohabitation statuses because there are often different outcomes for the health and well-being of pregnant women and their children in different settings. Further prospective cohort research is needed to investigate the social aspects that link marital transition and dietary intake outcomes among pregnant women in sub-Saharan African countries.

Parallel with globalisation, pronounced changes in the human behaviour and lifestyle such as decreased consumption of fruits and vegetables and increased consumption of unhealthy foods [38], have resulted in escalating rates of overweight and obesity among pregnant women. The trends of overweight and obesity among pregnant women in Tanzania has changed from being considered as a mild disorder to the major causes of morbidity and mortality associated with non-communicable diseases [13]. According to MUAC measures, this study found that only twenty one pregnant women had MUAC measures above 33cm [25]. Further, our study revealed that out of 420 pregnant women, only 12.6% and 29.3% consumed at least four servings of fruits and vegetables per week respectively. Similar findings were documented in the previous studies from low-income countries, where overweight and obesity pregnant women consumed less vegetable and fruits [12, 39]. In multivariate analysis, our data revealed that pregnant women who were overweight or obesity were significantly associated with poor dietary intake. Pregnant women with overweight and obesity need nutrition counseling that are supported by scientific evidence and that can be easily understood and translated into everyday life to improve maternal and birth outcomes. Findings from large, long-term randomised controlled trials provide convincing evidence that changes made in physical activity levels and dietary habits are effective in delaying, and possibly preventing, progression from overweight and obesity to non-communicable diseases [40]. Future prospective cohort research is needed to investigate the link between obesity and dietary patterns among pregnant women.

This study also highlighted an important finding, the protective effect to poor dietary intake among pregnant women who reported taking SP tablets. In Tanzania, SP tablets are offered to all pregnant women attending antenatal clinics between 16 and 24 weeks and, between 28 and

32 weeks [41]. The linkage between taking SP tablets and poor dietary intake protection could not established because malaria and nutrition interventions are well integrated into antenatal care in Tanzania, and both impart pregnant women with essential knowledge [42]. Further studies are needed to broaden the understanding on this relationship and, help researchers in sub-Saharan African countries to develop tailored interventions to improve maternal and birth outcomes.

## Strengths and limitations

There are several strengths of this study. First, the study was able to use reliable survey data and blood samples from a large population sample and measured diet during the pregnancy using both the Prime Diet Quality Score tool and 24-hour dietary recall. Second, this study provides important information on factors associated with dietary intake among pregnant women of gestation period less than 28 weeks, which is important for fetal development, given the rapid cell growth, and development of immune cells and organs in the first trimester [43, 44]. Third, the present study is that it links social and biological data with dietary data and allows analysis of dietary intake. However, there were several limitations of the study. First, we inevitably have some level of measurement error in both dietary and social and biological data, as both were based on self-report. This source of error is, however, expected to be largely random, producing valid estimates for the study population. Second, we derived PDQS scores from 24-hour recalls, and there were limited precedents in published literature for converting these scores to equivalent scores for the food frequency questionnaire (FFQ). The validity of using the PDQS score assessed using 24-hour recall is an area of active research. Notably, the 24-hour recall method is used widely in developing countries and our findings provide support for the use of this metric for deriving PDQS in these settings. Our findings may not be generalisable to populations where dietary patterns and determinants outcomes differ from rural Tanzania. Associations may be stronger in populations with more prevalent micronutrients and other deficiencies in pregnant women.

## Conclusions

The results of this study affirm that cohabitation and obesity affect dietary intake among pregnant women differently compared to marriage in rural settings of Tanzania. Further, the findings suggest that public health action is needed to promote the consumption of fruits and vegetables among pregnant women in Mbeya. We recommend prospective cohort research to investigate the social aspects that link poor dietary intake outcomes for developing a tailored gestational intervention to improve maternal and birth outcomes in sub-Saharan African countries.

## Supporting information

**S1 Checklist. STROBE statement—Checklist of items that should be included in reports of observational studies.**
(DOCX)

## Author Contributions

**Conceptualization:** Erick Killel, Nyamizi Ngasa, Adam Hancy, Tedson Lukindo, Ramadhan Noor, Abraham Sanga, Germana H. Leyna, Ray M. Masumo.

**Data curation:** Adam Hancy, Tedson Lukindo.

**Formal analysis:** Erick Killel, Geofrey Mchau, Hamida Mbilikila, Kaunara Azizi, Nyamizi Ngasa, Adam Hancy, Tedson Lukindo, Ramadhan Mwiru, Ray M. Masumo.

**Investigation:** Erick Killel, Kaunara Azizi, Adam Hancy, Tedson Lukindo, Ramadhan Noor, Abraham Sanga.

**Methodology:** Erick Killel, Hamida Mbilikila, Kaunara Azizi, Nyamizi Ngasa, Adam Hancy, Tedson Lukindo, Ramadhan Noor, Ray M. Masumo.

**Project administration:** Erick Killel, Geofrey Mchau, Tedson Lukindo, Ramadhan Noor, Abraham Sanga, Patrick Codjia.

**Supervision:** Ramadhan Mwiru, Ramadhan Noor, Patrick Codjia, Germana H. Leyna, Ray M. Masumo.

**Validation:** Adam Hancy.

**Writing – original draft:** Erick Killel, Geofrey Mchau, Hamida Mbilikila, Kaunara Azizi, Nyamizi Ngasa, Tedson Lukindo, Ramadhan Mwiru, Abraham Sanga, Patrick Codjia, Germana H. Leyna, Ray M. Masumo.

**Writing – review & editing:** Geofrey Mchau, Nyamizi Ngasa, Ramadhan Mwiru, Abraham Sanga, Patrick Codjia, Germana H. Leyna, Ray M. Masumo.

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
