## [Decision Letter · Decision Letter 0]

12 Sep 2023

PGPH-D-23-01270

Factors associated with poor dietary intake among pregnant women in Mbeya, Tanzania

Dear Dr. Masumo,

Thank you for submitting your manuscript to PLOS Global Public Health. After careful consideration, we feel that it has merit but does not fully meet PLOS Global Public Health’s publication criteria as it currently stands. Therefore, we invite you to submit a revised version of the manuscript that addresses the points raised during the review process.

EDITOR: Please insert comments here and delete this placeholder text when finished. Be sure to:

Please ensure that your decision is justified on PLOS Global Public Health’s publication criteria and not, for example, on novelty or perceived impact.

We look forward to receiving your revised manuscript.

Kind regards,

Dickson Abanimi Amugsi, PhD

Academic Editor

Journal Requirements:

1. We noticed you have some minor occurrence of overlapping text with the following previous publication(s), which needs to be addressed:

- https://link.springer.com/content/pdf/10.1186/s13104-016-2122-3.pdf

- https://journals.plos.org/globalpublichealth/article?id=10.1371%2Fjournal.pgph.0001828

- https://pureadmin.qub.ac.uk/ws/files/195797445/Association_between_oral_health_status_and_future_dietary_intake_and_diet_quality_in_older_men_the_PRIME_study.pdf

In your revision ensure you cite all your sources (including your own works), and quote or rephrase any duplicated text outside the methods section. Further consideration is dependent on these concerns being addressed."""

Additional Editor Comments (if provided):

Reviewers' comments:

Reviewer's Responses to Questions

**Comments to the Author**

1. Does this manuscript meet PLOS Global Public Health’s publication criteria? Is the manuscript technically sound, and do the data support the conclusions? The manuscript must describe methodologically and ethically rigorous research with conclusions that are appropriately drawn based on the data presented.

Reviewer #1: Yes

Reviewer #2: Yes

Reviewer #3: No

2. Has the statistical analysis been performed appropriately and rigorously?

Reviewer #1: Yes

Reviewer #2: Yes

Reviewer #3: Yes

3. Have the authors made all data underlying the findings in their manuscript fully available (please refer to the Data Availability Statement at the start of the manuscript PDF file)?

Reviewer #1: Yes

Reviewer #2: Yes

Reviewer #3: Yes

4. Is the manuscript presented in an intelligible fashion and written in standard English?

Reviewer #1: Yes

Reviewer #2: Yes

Reviewer #3: No

5. Review Comments to the Author

Reviewer #1: The article meets the criteria for PLOS Global public health journal publication and has a technically sound manuscript with conclusions that are supported by data presented in the results section. The cross sectional study design for the primary research question posed projects an ethically and methodologically sound design with mentioned limitations and weaknesses. This is further elaborated by the drawn conclusions which even with the mentioned biases in the design, answers the question posed and goes on to identify the associated protective and detrimental influencers to good dietary practices of pregnant women in Mbeya, Tanzania. Despite the mentioned practical hindrances to screening of pregnant women for their dietary practices (a vital public health problem if i might add) in a developing nation, the conclusions also justify the need for national dietary guidelines to guide nutrition during pregnancy for Tanzanian women.

An astounding attempt at statistical analysis of the data is seen through the sub-categorization of data captured from the questionnaire and attempt to find associations and correlations between the data and clear and correct use of mathematical formulas to infer or defer any data correlation or association. Additionally, data availability seems adequate through data projected in the tables and despite an isolated grammatical error the manuscript is presented in good palatable english. Line 143- grammar error (some text missing)

However, editorial recommendations are listed below.

line 123 &126/7- Provide a rationale and justification for exclusion criteria e.g why cut off at 28 weeks gestational age when the title of the article includes ‘pregnant women’ and why exclude pregnant woman taking medication and define the criteria used to define medication as shall be highlighted later in the review.

Line 144- provide a rationale for use of a screening tool (PDQS) that has not been tested in the population group of interest which is pregnant women in low to middle income countries (provide evidence if the case is otherwise). This is because even though the authors acknowledge the limitations of their choice of screening tool (PDQS in this case), the identification of the ‘at risk’ pregnant women is highly dependent on the validity, sensitivity and specificity of this screening tool and conclusions made should be translated and synthesized after acknowledging the strengths and weaknesses of the screening tool and why it was used despite the identified/known strengths/weakness.This critical analysis of the tool can be analyzed in a thorough and similar manner to the method of its translation to KiSwahili which appear to have been executed in an excellent manner.

line 163- The recall bias associated with a 24 hr recall has be acknowledged and a 30 day recall might have brought a different comparable outcome that could also be acknowledged in an obvious attempt to include a wider dietary assessment but this is speculation from a reviewer who shares the opinion that acknowledging such gaps might strengthen the recommendations for future research practice.

Line 267- Elaborate on the inclusion of pregnant women taking Fansidar(line 267,278, 336), iron and folic acid (line270) (chemicals that the reviewer considers to be medication) when the exclusion criteria excluded women taking medication.

285- The discussion appears to underreport the relationship between the socioeconomic environment or disease states like anemia, hypoalbuminemia etc to dietary intake even though such indices were reported to have been captured in the method and results. As a reader i remain unquenched as to whether the relationship was not reported because the relationship was insignificant to report or it was simple neglect.

Reviewer #2: Comment: we do not assess poor dietary intake, rather dietary intake to ascertain if its poor or not

in the methods: concern on authors assessing {“poor diet”) poor diet is an outcome of dietary assessment. You can only determine of the respondents consumed poor diet or good diet after the assessment. It is a product of assessment. I am not sure there is a tool designed to assess poor diet. We have tools to assess dietary intake. Analysis will confirm if the diet was poor or not. If the authors have to focus on poor diets, then they need to state of there are other findings that poor diets in the study population that has led to this study to assess the factors contributing to poor intake. However, if there is no primary data on this, then the focus should be to assess dietary intake.

There is need to review the statements in the whole manuscript, e.g. the statement “A cross-sectional study design was conducted in Mbeya” should delete the word design

The refined grains are portrayed as a good thing while in essence the focus of assessment should be the proportion of mothers consuming unrefined grains.

Poor dietary intake has seriously detrimental consequences for pregnancy and born children both in developed and developing countries (this statement has been overused, in the abstract, introduction and discussions)

In the discussion, the comparison between pregnant women and women with HIV may not be the ideal. Persons with HIV tend to receive a lot of nutrition education to improve their nutrition status given the effects of poor nutrition on HIV outcomes.

The findings and the title need to be aligned better; the results have had very specific association cohabiting, obesity, and malaria treatment. Maybe the title should consider capturing this. From the findings cohabiting did promote good dietary intake, it therefore does not qualify to be a factor of poor dietary intake.

The title suggests factors associated with poor dietary intake. Since the title has a bias on the poor intake, then emphasis on this should be discussed. It would be good to highlight possible factors that make cohabiting promote good dietary intake, this is not clear.

I also not only 5 % of the women were obese/overweight, but that’s also quite a small proportion and wondering if this is sufficient to draw the conclusion on effects of obesity on poor dietary intake. What is the link, a review of more literature for the discussions.

Fansider intake seems to have promoted dietary intake, however authors should discuss why this is so. How does malaria treatment improve dietary intake ?

The only dietary items focused on in the results are vegetables and fruits and partly grains, why haven’t the authors mentioned the other foods? Were they not of interest to pregnant women especially proteins (animal sources) given the nutrient needs in pregnancy? Especially iron?

Reviewer #3: Dear authors, thank you for the opportunity to review your work. This study has the potential to be a very interesting and valuable study, but some major modifications are required as outlined below:

Language editing is required. I started to correct some of the grammatical errors, but it became evident that there are many errors that will require language editing to fix these.

Introduction:

Gives a broad overview of the topic, however, language issues prevented from being able to fully understand what the authors are trying to explain.

Methods:

Study design: Adequately described.

Study population:

Please see further comments on manuscript provided.

Sample size and sampling procedure:

Adequately described.

Data collection:

Dietary intake and diet quality assessment:

More detail on the exact process followed to collect dietary intake data is required. Please see manuscript provided for more feedback.

I am also concerned about the relevance of using a tool developed for coronary heart disease risk when numerous other tools/instruments are available to determine dietary quality in pregnancy.

Demographic and socio-economic factors:

Adequately described.

Discuss anthropometric measurements under a separate heading. Also, please indicate which standardised techniques and/or references were used for measuring anthropometry.

Laboratory assessment:

Adequately described.

Data analysis:

Please indicate what cut-off level was used when reporting on P-values.

Ethics statement:

Please indicate whether informed assent was obtained from minors.

Results:

Reliability: Adequately described.

The characteristics of the study population: Adequately described.

Dietary intake and diet quality:

What was the mean/median PDQS score and how many women in the total sample were classified as having poor PDQS score?

This entire section (lines 258 - 265) needs to be rephrased since the text does not clearly explain what is indicated in Table 2.

Sentence in lines 261 - 264 needs to be rephrased as it does not make sense currently.

Lines 264 - 265: "Refined grains, baked goods and sugar-sweetened beverages were the most commonly consumed unhealthy foods" - According to the data presented in Table 2, 6.2% of women consumed four servings or more from the sugar sweetened beverages group per week? Thus it does not make sense how you make this statement?

Bivariate analysis:

Adequately described. Please make sure of P-values, since the P-value for those who received Fansidar tables during pregnancy is indicated as 0.077 in the table, but in the text it is flagged as being significant? It will also be good to explain (either in the introduction or discussion) what these tablets are given for and how often.

Multivariate analysis:

Again, please verify what cut-off level/value was used when reporting on P-values.

No P-values indicated in Table 4, thus the reference to P-value in line 284 does not make sense.

Discussion:

Needs language editing.

Also consider differences on dietary assessment tools used ad possible reasons for differences between findings from the current study and that of others (line 294).

Lines 333-344: Regarding IPT needs to be discussed in context of the findings of the current study.

6. PLOS authors have the option to publish the peer review history of their article (what does this mean?). If published, this will include your full peer review and any attached files.

**Do you want your identity to be public for this peer review?** For information about this choice, including consent withdrawal, please see our Privacy Policy.

Reviewer #1: **Yes: **Benson Tarisai Gombe

Reviewer #2: No

Reviewer #3: No

---

## [Decision Letter · Decision Letter 1]

3 Nov 2023

PGPH-D-23-01270R1

Dietary intake and associated risk factors among pregnant women in Mbeya, Tanzania

Dear Dr. Masumo,

Thank you for submitting your manuscript to PLOS Global Public Health. After careful consideration, we feel that it has merit but does not fully meet PLOS Global Public Health’s publication criteria as it currently stands. Therefore, we invite you to submit a revised version of the manuscript that addresses the points raised during the review process.

EDITOR: Please insert comments here and delete this placeholder text when finished. Be sure to:

Please ensure that your decision is justified on PLOS Global Public Health’s publication criteria and not, for example, on novelty or perceived impact.

We look forward to receiving your revised manuscript.

Kind regards,

Dickson Abanimi Amugsi, PhD

Academic Editor

Journal Requirements:

Additional Editor Comments (if provided):

Reviewers' comments:

Reviewer's Responses to Questions

**Comments to the Author**

1. If the authors have adequately addressed your comments raised in a previous round of review and you feel that this manuscript is now acceptable for publication, you may indicate that here to bypass the “Comments to the Author” section, enter your conflict of interest statement in the “Confidential to Editor” section, and submit your "Accept" recommendation.

Reviewer #1: (No Response)

Reviewer #3: All comments have been addressed

2. Does this manuscript meet PLOS Global Public Health’s publication criteria? Is the manuscript technically sound, and do the data support the conclusions? The manuscript must describe methodologically and ethically rigorous research with conclusions that are appropriately drawn based on the data presented.

Reviewer #1: Yes

Reviewer #3: Yes

3. Has the statistical analysis been performed appropriately and rigorously?

Reviewer #1: Yes

Reviewer #3: Yes

4. Have the authors made all data underlying the findings in their manuscript fully available (please refer to the Data Availability Statement at the start of the manuscript PDF file)?

Reviewer #1: Yes

Reviewer #3: Yes

5. Is the manuscript presented in an intelligible fashion and written in standard English?

Reviewer #1: Yes

Reviewer #3: Yes

6. Review Comments to the Author

Reviewer #1: Line267-I remain unanswered by the lack of a rationale or clarity towards the reported exclusion of pregnant women taking medication whilst there was inclusion of pregnant women taking fansidar, iron supplements etc which by definition can fall under the category of medication.Kindly clarify this antagonistic exclusion criteria.

Reviewer #3: Dear authors,

Thank you for considering and implementing the changes recommended during the previous revision.

Only a few minor language / technical changes still required:

Results section:

Lines 281 - 282: Please revise this sentence and rephrase to perhaps read as follows: "However, refined grains and baked goods represented the highest percentage of servings consumed per week out of the six unhealthy foods assessed."

Discussion section:

Lines 316 - 319: Please revise this sentence and consider rephrasing to: "Guidelines of other countries like UK and USA recommend a plant-based diet, rich in fruit, vegetables...."

Line 349: the word "Finding" should rather be "Findings"

Line 353: The word "obese" should rather be "obesity"

Line 356: Insert the word "an" between the words "highlighted" and "finding"

7. PLOS authors have the option to publish the peer review history of their article (what does this mean?). If published, this will include your full peer review and any attached files.

**Do you want your identity to be public for this peer review?** For information about this choice, including consent withdrawal, please see our Privacy Policy.

Reviewer #1: **Yes: **Benson Gombe

Reviewer #3: No

---

## [Decision Letter · Decision Letter 2]

14 Nov 2023

PGPH-D-23-01270R2

Dietary intake and associated risk factors among pregnant women in Mbeya, Tanzania

Dear Dr. Masumo,

Thank you for submitting your manuscript to PLOS Global Public Health. After careful consideration, we feel that it has merit but does not fully meet PLOS Global Public Health’s publication criteria as it currently stands. Therefore, we invite you to submit a revised version of the manuscript that addresses the points raised during the review process.

EDITOR: Please insert comments here and delete this placeholder text when finished. Be sure to:

Please ensure that your decision is justified on PLOS Global Public Health’s publication criteria and not, for example, on novelty or perceived impact.

We look forward to receiving your revised manuscript.

Kind regards,

Dickson Abanimi Amugsi, PhD

Academic Editor

Journal Requirements:

Additional Editor Comments (if provided):

Thank you for your quick action on the reviewers' comments. Reviewer 1 felt the issue they raised regarding your exclusion criteria was not adequately addressed. Please address this concern and return the manuscript to me for a final decision.

Thank you.

Reviewers' comments:

Reviewer's Responses to Questions

**Comments to the Author**

1. If the authors have adequately addressed your comments raised in a previous round of review and you feel that this manuscript is now acceptable for publication, you may indicate that here to bypass the “Comments to the Author” section, enter your conflict of interest statement in the “Confidential to Editor” section, and submit your "Accept" recommendation.

Reviewer #1: (No Response)

Reviewer #3: All comments have been addressed

2. Does this manuscript meet PLOS Global Public Health’s publication criteria? Is the manuscript technically sound, and do the data support the conclusions? The manuscript must describe methodologically and ethically rigorous research with conclusions that are appropriately drawn based on the data presented.

Reviewer #1: Partly

Reviewer #3: Yes

3. Has the statistical analysis been performed appropriately and rigorously?

Reviewer #1: Yes

Reviewer #3: Yes

4. Have the authors made all data underlying the findings in their manuscript fully available (please refer to the Data Availability Statement at the start of the manuscript PDF file)?

Reviewer #1: Yes

Reviewer #3: Yes

5. Is the manuscript presented in an intelligible fashion and written in standard English?

Reviewer #1: Yes

Reviewer #3: Yes

6. Review Comments to the Author

Reviewer #1: The exclusion criteria remains unchanged, it mentions that pregnant women taking medication were excluded and fails to explain why if this was so, there was inclusion of pregnant women taking fansidar(chemoprophylaxis for malaria), iron and folate supplements. All these chemicals are also medication so this antagonistic statement remains an issue.

Recommendations would advice to provide a meaningful exclusion criteria eg, the study excluded pregnant women taking medication for other reasons except malaria chemoprophylaxis plus iron and folate supplements

Reviewer #3: Dear authors,

Thank you for making the necessary / suggested changes to the manuscript. This research represents a very relevant topic of research and it is recommended that the manuscript be accepted for publication.

7. PLOS authors have the option to publish the peer review history of their article (what does this mean?). If published, this will include your full peer review and any attached files.

**Do you want your identity to be public for this peer review?** For information about this choice, including consent withdrawal, please see our Privacy Policy.

Reviewer #1: No

Reviewer #3: No

---

## [Editor Report · Decision Letter 3]

28 Nov 2023

Dietary intake and associated risk factors among pregnant women in Mbeya, Tanzania

PGPH-D-23-01270R3

Dear Dr Masumo,

We are pleased to inform you that your manuscript 'Dietary intake and associated risk factors among pregnant women in Mbeya, Tanzania' has been provisionally accepted for publication in PLOS Global Public Health.

Best regards,

Dickson Abanimi Amugsi, PhD

Academic Editor